# Constraining the Viscous Dark Energy Equation of State in $f(R, L_m)$ Gravity

Lakhan V. Jaybhaye, Raja Solanki, Sanjay Mandal and Pradyumn Kumar Sahoo *

Department of Mathematics, Birla Institute of Technology and Science-Pilani, Hyderabad Campus,
Hyderabad 500078, India; p20200477@hyderabad.bits-pilani.ac.in (L.V.J.);
p20200003@hyderabad.bits-pilani.ac.in (R.S.); p20180444@hyderabad.bits-pilani.ac.in (S.M.)
* Correspondence: pksahoo@hyderabad.bits-pilani.ac.in

**Abstract:** In this article, we attempt to describe the cosmic late-time acceleration of the universe in the framework of $f(R, L_m)$ gravity, by using an effective equation of state, when bulk viscosity is taken into account. We presume a non-linear $f(R, L_m)$ functional form, specifically, $f(R, L_m) = \frac{R}{2} + L_m^\alpha$, where $\alpha$ is a free model parameter. We obtain the exact solution of our bulk viscous matter dominated $f(R, L_m)$ model, and then we utilize the combined $H(z)$ + Pantheon + Analysis datasets to estimate the best fit values of the free parameters of our model. Then, we characterize the behavior of the matter–energy density, effective pressure, and the equation of state (EoS) parameter, incorporating the viscous type fluid. The evolution profile of the effective EoS parameter depicts an acceleration phase of the cosmic expansion, whereas the pressure, with the effect of viscosity, exhibits negative behavior, that can lead to the accelerating expansion of the universe. Moreover, the cosmic matter–energy density shows the expected positive behavior. Further, we investigate the behavior of the statefinder parameters for the assumed $f(R, L_m)$ model. We find that the evolutionary trajectory of the given model lies in the quintessence region. In addition, we employ the Om diagnostic test, that indicates that our model exhibits quintessence behavior. Lastly, we check the energy condition criteria and find that the violation of SEC occurs in the past, whereas NEC and DEC satisfy the positivity criteria. We find that our $f(R, L_m)$ cosmological model, with the effect of bulk viscosity, provides a good fit of the recent observational data and can efficiently describe the cosmic expansion scenario.

**Keywords:** $f(R, L_m)$ gravity; bulk viscosity; equation of state parameter; observational datasets; statefinder parameter; Om diagnostic

## 1. Introduction

Cosmology faced a dramatic change when the observational evidence from type Ia supernovae searches [1,2] confirmed the accelerating behavior of cosmic expansion. In the last two decades, a plethora of observational results, such as the large-scale structure [3], the Wilkinson Microwave Anisotropy Probe [4], the cosmic microwave background radiation [5,6], and baryonic acoustic oscillations [7,8], have agreed with the observed cosmic acceleration. The most prominent explanation to describe this accelerating scenario, is the presence of a dark energy component, characterized by an equation of state $\omega = -1.018 \pm 0.057$, for a flat universe [9]. Another promising way to describe the accelerating expansion of the universe, by bypassing the undetected dark energy component, is to consider that the more generic action describes the gravitational field. The cosmological models in which the Einstein–Hilbert action of general relativity (GR) is modified by introducing the generic function $f(R)$, where $R$ denotes the Ricci scalar curvature, first proposed in [10–12]. The $f(R)$ gravity model is capable of describing the expansion mechanism without invoking any exotic dark energy component [13,14]. Observational signatures of $f(R)$

gravity models, along with the solar system and equivalence principle constraints, are presented in [15–17]. In the context of solar system tests, viable cosmological models of $f(R)$ gravity do exist [18–20]. Odintsov et al. have analyzed the $H_0$ tension and the role of energy conditions in $f(R)$ gravity models [21,22]. One can follow the references [23–29] to see the various implications of cosmological models of $f(R)$ gravity.

A generalization of the curvature based $f(R)$ gravity, that incorporates an explicit coupling of the generic function $f(R)$ with the matter Lagrangian density $L_m$, appeared in [30]. This coupling case was further extended to the case of arbitrary matter geometry couplings [31]. Harko and Lobo investigated the curvature–matter couplings in modified gravity, from linear aspects to conformally invariant theories [32]. Models with non-minimal matter geometry couplings have great astrophysical and cosmological implications. Harko studied the galactic rotation curves, the matter Lagrangian and the energy momentum tensor, thermodynamical features, coupling matter and curvature in Weyl geometry, in the context of non-minimal couplings [33–36]. Moreover, Bertolami et al. [37] investigated curvature–matter couplings in modified gravity, and Faraoni examined the viability criterion for modified gravity with an extra force [38]. Further, Harko and Lobo recently proposed [39] $f(R, L_m)$ gravity theory, that is a generalization of matter–curvature coupling theories, where $f(R, L_m)$ is a generic function that depends on the Ricci scalar $R$ and the matter Lagrangian $L_m$. In this theory, the covariant divergence of the stress–energy tensor does not vanish, an extra force, orthogonal to four velocities, arises, and the motion of the test particle is non-geodesic. The models of $f(R, L_m)$ gravity theory disobey the equivalence principle, and that is restrained by the solar system experimental tests [40,41]. Recently, several interesting results on $f(R, L_m)$ gravity have appeared, for instance, see references [42–45].

In the presented manuscript, we explore the cosmological $f(R, L_m)$ model that exhibits a viscous type fluid. The introduction of the coefficient of viscosity in cosmology models has a long history. From a hydrodynamicist's point of view, there are two viscosity coefficients commonly presented in the literature, namely the bulk viscosity coefficient $\zeta$ and the shear viscosity coefficient $\eta$. By assuming the, observationally supported, spatial isotropy of the cosmos, shear viscosity can be omitted. Whenever a system deviates from its thermal equilibrium, then, to recover its thermal equilibrium state, an effective pressure is generated. Bulk viscosity in a cosmological fluid is the manifestation of such an effective pressure. The idea is to consider the bulk viscosity coefficient $\zeta$ in the $f(R, L_m)$ gravity model. We assume that the coefficient of bulk viscosity $\zeta$ satisfies a scaling law, and that it reduces the Einstein case to a form proportional to the Hubble parameter. It has been shown that this scaling law is quite useful. Viscous fluid cosmological models are further discussed in [46–53].

The present manuscript is organized in the following manner. In Section 2, we present the action and basic formulation governing the dynamics in $f(R, L_m)$ gravity. In Section 3, we present the Friedmann-like equations corresponding to the flat FLRW universe. In Section 4, we assume an $f(R, L_m)$ functional and then we calculate the expression for the Hubble parameter and the equation of state (EoS) parameter, relating the pressure term of bulk viscous matter with its energy density. In Section 5, we estimate the values of the $H_0$ and model parameters that are consistent with observations, by incorporating the combined $H(z)$ + Pantheon + Analysis datasets. In addition, we characterize the behavior of various parameters such as density, effective pressure, and the EoS parameter. Further, in Section 6, we investigate the $r$–$s$ parameter trajectory of our $f(R, L_m)$ model, to check the dark energy behavior recognized by the assumed model. Moreover, in Section 7 and Section 8, we employ the Om diagnostic test and energy condition criteria. Finally, in Section 9, we conclude our findings.

## 2. $f(R, L_m)$ Gravity Theory

The generic action for $f(R, L_m)$ gravity reads as

$$S = \int f(R, L_m) \sqrt{-g} d^4x \tag{1}$$

Here, $R$ represents the Ricci scalar curvature and $L_m$ denotes the matter Lagrangian. One can obtain the Ricci scalar $R$ by contracting the Ricci tensor $R_{\mu\nu}$ as

$$R = g^{\mu\nu} R_{\mu\nu} \tag{2}$$

where the Ricci tensor is given by

$$R_{\mu\nu} = \partial_\lambda \Gamma^\lambda_{\mu\nu} - \partial_\mu \Gamma^\lambda_{\lambda\nu} + \Gamma^\lambda_{\mu\nu} \Gamma^\sigma_{\sigma\lambda} - \Gamma^\lambda_{\nu\sigma} \Gamma^\sigma_{\mu\lambda} \tag{3}$$

with $\Gamma^\alpha_{\beta\gamma}$ representing the components of the Levi–Civita connection.

Now, we obtain the following field equation, governing the dynamics of gravitational interactions, by varying the action (1) with respect to the metric tensor $g_{\mu\nu}$,

$$f_R R_{\mu\nu} + (g_{\mu\nu}\Box - \nabla_\mu \nabla_\nu)f_R - \frac{1}{2}(f - f_{L_m} L_m)g_{\mu\nu} = \frac{1}{2} f_{L_m} T_{\mu\nu} \tag{4}$$

Here, $f_R \equiv \frac{\partial f}{\partial R}$, $f_{L_m} \equiv \frac{\partial f}{\partial L_m}$, and $T_{\mu\nu}$ represents the stress–energy tensor for the cosmic fluid, defined by

$$T_{\mu\nu} = \frac{-2}{\sqrt{-g}} \frac{\delta(\sqrt{-g}L_m)}{\delta g^{\mu\nu}} \tag{5}$$

The connection among the energy–momentum scalar $T$, the matter Lagrangian term $L_m$, and the Ricci scalar curvature $R$ acquired by contracting the field, Equation (4), is

$$R f_R + 3\Box f_R - 2(f - f_{L_m} L_m) = \frac{1}{2} f_{L_m} T \tag{6}$$

Here, $\Box F = \frac{1}{\sqrt{-g}} \partial_\alpha (\sqrt{-g} g^{\alpha\beta} \partial_\beta F)$ for any scalar function $F$.

In addition, one can obtain the following relation by employing the covariant derivative in Equation (4)

$$\nabla^\mu T_{\mu\nu} = 2\nabla^\mu ln(f_{L_m}) \frac{\partial L_m}{\partial g^{\mu\nu}} \tag{7}$$

## 3. Motion Equations in $f(R, L_m)$ Gravity

In order to probe the cosmological implications, we consider the following homogeneous and spatially isotropic FLRW metric [54]

$$ds^2 = -dt^2 + a^2(t)[dx^2 + dy^2 + dz^2] \tag{8}$$

where, $a(t)$ is the cosmic scale factor. The Ricci scalar obtained for metric (8) is

$$R = 6\frac{\ddot{a}}{a} + 6\left(\frac{\dot{a}}{a}\right)^2 = 6(\dot{H} + 2H^2) \tag{9}$$

where $H = \frac{\dot{a}}{a}$ is the Hubble parameter.

The energy–momentum tensor, comprising the energy density $\rho$ and the pressure $\bar{p}$ of the cosmic fluid with the viscosity effect, is given by,

$$\mathcal{T}_{\mu\nu} = (\rho + \bar{p})u_\mu u_\nu + \bar{p} g_{\mu\nu} \tag{10}$$

where $\bar{p} = p - 3\zeta H$ and $u^\mu = (1, 0, 0, 0)$ are components of the four velocities. Here, $p$ is the usual pressure and $\zeta > 0$ is the coefficient of bulk viscosity.

The connection between matter–energy density and the usual pressure, is given as [55]

$$p = (\gamma - 1)\rho \tag{11}$$

where $\gamma$ is a constant with $0 \le \gamma \le 2$. Hence, the effective equation of state characterizing the bulk viscous cosmic fluid reads as [56–58]

$$\bar{p} = (\gamma - 1)\rho - 3\zeta H \tag{12}$$

Under the constraint of homogeneity and spatial isotropy, the cosmic fluid incorporating viscosity possesses dissipative phenomenon. Considering viscosity in a cosmic fluid can minimize the ideal characteristics of a fluid, and contributes negatively to the total pressure. This is presented in [59–61].

The Friedmann equations, that characterize the bulk viscous matter dominated universe in $f(R, L_m)$ gravity, read as [62]

$$3H^2 f_R + \frac{1}{2}(f - f_R R - f_{L_m} L_m) + 3H\dot{f}_R = \frac{1}{2}f_{L_m}\rho \tag{13}$$

and

$$\dot{H}f_R + 3H^2 f_R - \ddot{f}_R - 3H\dot{f}_R + \frac{1}{2}(f_{L_m}L_m - f) = \frac{1}{2}f_{L_m}\bar{p} \tag{14}$$

## 4. Cosmological $f(R, L_m)$ Model

We choose the following $f(R, L_m)$ function in order to explore the dynamics of the universe that possesses viscosity [62,63],

$$f(R, L_m) = \frac{R}{2} + L_m^\alpha \tag{15}$$

Here, $\alpha$ is a free model parameter. The model under consideration is more general in nature and it is motivated by the functional form $f(R, L_m) = f_1(R) + f_2(R)G(L_m)$, that represents arbitrary matter–geometry coupling [63].

Then, for this specific functional form, with $L_m = \rho$ [64], the Friedmann Equations (13) and (14) characterizing the universe dominated by bulk viscous matter become

$$3H^2 = (2\alpha - 1)\rho^\alpha \tag{16}$$

and

$$2\dot{H} + 3H^2 = \left\{(\alpha - 1)\rho - \alpha\bar{p}\right\}\rho^{\alpha-1} \tag{17}$$

Now, by using Equation (7), we obtain the following matter conservation equation for our bulk viscous cosmological $f(R, L_m)$ model

$$(2\alpha - 1)\dot{\rho} + 3\gamma H\rho = 0 \tag{18}$$

From Equations (16) and (17), one obtains

$$\dot{H} + \frac{3\alpha\gamma}{2(2\alpha - 1)}H^2 = \frac{3}{2}\left(\frac{3}{2\alpha - 1}\right)^{\frac{\alpha-1}{\alpha}}\alpha\zeta H^{\frac{3\alpha-2}{\alpha}} \tag{19}$$

We substitute $\frac{1}{H}\frac{d}{dt} = \frac{d}{dln(a)}$, so that Equation (19) becomes

$$\frac{dH}{dln(a)} + \frac{3\alpha\gamma}{2(2\alpha-1)}H = \frac{3}{2}\left(\frac{3}{2\alpha-1}\right)^{\frac{\alpha-1}{\alpha}}\alpha\zeta H^{\frac{2(\alpha-1)}{\alpha}} \tag{20}$$

On integrating Equation (20), we obtain the expression for the Hubble parameter, as follows

$$H(z) = \left\{H_0^{\frac{2-\alpha}{\alpha}}(1+z)^{\frac{3\gamma(2-\alpha)}{2(2\alpha-1)}} + \frac{3\zeta}{\gamma}\left(\frac{2\alpha-1}{3}\right)^{\frac{1}{\alpha}}[1-(1+z)^{\frac{3\gamma(2-\alpha)}{2(2\alpha-1)}}]\right\}^{\frac{\alpha}{2-\alpha}} \tag{21}$$

where $H(0) = H_0$ represents the present value of the Hubble parameter. In particular, for the case $\alpha = 1$, with $\gamma = 1$ and $\zeta = 0$, the solution reduces to $H(z) = H_0(1+z)^{\frac{3}{2}}$, the usual ordinary matter dominated universe.

The effective equation of state parameter for our bulk viscous cosmological model is given by

$$\omega_{eff} = \frac{p_{eff}}{\rho} = \gamma - 1 - \frac{3\zeta H}{\rho} \tag{22}$$

By using Equations (16) and (21), one can acquire

$$\omega_{eff} = \gamma - 1 - 3\zeta\left(\frac{2\alpha-1}{3}\right)^{\frac{1}{\alpha}}\left\{H_0^{\frac{2-\alpha}{\alpha}}(1+z)^{\frac{3\gamma(2-\alpha)}{2(2\alpha-1)}} + \frac{3\zeta}{\gamma}\left(\frac{2\alpha-1}{3}\right)^{\frac{1}{\alpha}}[1-(1+z)^{\frac{3\gamma(2-\alpha)}{2(2\alpha-1)}}]\right\}^{-1} \tag{23}$$

## 5. Data, Methodology, and Physical Interpretation

In this section, we estimate the parameter values of our model, that are appropriate to describe the various cosmic epochs, by invoking the $H(z)$ and Pantheon + Analysis datasets. To calculate the suitable values of $H_0$ and the model parameters $\alpha$, $\gamma$, and $\zeta$, we incorporate 31 points of the $H(z)$ dataset and 1701 points from the Pantheon + Analysis samples. To estimate the mean values of the parameters of our viscosity model, we apply the Bayesian technique and likelihood function, along with the Markov chain Monte Carlo (MCMC) method, in `emcee` in the Python library [65].

### 5.1. H(z) Dataset

It is well known, that the Hubble parameter can directly investigate cosmic expansion. In terms of redshift, the Hubble parameter can be acquired as $H(z) = -\frac{1}{1+z}\frac{dz}{dt}$. Since $dz$ is derived from spectroscopic surveys, therefore, one can obtain the model-independent $H(z)$ value by measuring $dt$. In this manuscript, we incorporate 31 data points of $H(z)$ measurements in the redshift range $0.07 \leq z \leq 2.41$ [66]. The complete list of 31 data points are in [67]. We define the chi-square function, to find out the mean values of the bulk viscous model parameters $\alpha$, $\gamma$, $\zeta$, and $H_0$ as follows,

$$\chi_H^2(H_0, \alpha, \gamma, \zeta) = \sum_{k=1}^{31}\frac{[H_{th}(z_k, H_0, \alpha, \gamma, \zeta) - H_{obs}(z_k)]^2}{\sigma_{H(z_k)}^2}. \tag{24}$$

Here, the theoretical value of the $H(z)$ acquired by our cosmological model, is represented by $H_{th}$, whereas $H_{obs}$ denotes its observed value and $\sigma_{H(z_k)}$ is the standard error.

### 5.2. Pantheon Dataset

Earlier, the observational results on type Ia supernovae confirmed that our universe is going through a phase of accelerated expansion. In the past two decades, observations on supernovae samples have increased dramatically. In 2018, 1048 samples of type Ia super-

novae, covering the redshift range $0.01 < z < 2.3$, were released, known as the Pantheon supernovae samples [68]. The PanSTARSS1 Medium, Deep Survey, SDSS, HST surveys, SNLS, and numerous low redshift surveys contribute to it. Recently, the Pantheon + Analysis sample, incorporating 1701 light curves of 1550 supernovae, in the redshift range $[0.001, 2.26]$, has been released [69]. The luminosity distance is taken to be [9],

$$
\begin{aligned}
D_L(z) &= \frac{c(1+z)}{H_0} S_k\left(H_0 \int_0^z \frac{1}{H(z')} dz'\right), \\
\text{where } S_k(x) &= \begin{cases} \sinh(x\sqrt{\Omega_k})/\Omega_k, & \Omega_k > 0 \\ x, & \Omega_k = 0 \\ \sin x\sqrt{|\Omega_k|})/|\Omega_k|, & \Omega_k < 0 \end{cases}
\end{aligned}
$$

For a spatially flat universe, we have

$$
D_L(z) = (1+z) \int_0^z \frac{c\,dz'}{H(z')}, \tag{25}
$$

where $c$ is the speed of light.

We have calculated the $\chi^2$ function for the Pantheon supernovae samples, by correlating the theoretical distance modulus

$$
\mu(z) = 5log_{10}D_L(z) + \mu_0, \tag{26}
$$

with

$$
\mu_0 = 5log(1/H_0 Mpc) + 25, \tag{27}
$$

such that

$$
\chi^2_{SN}(p_1, \ldots) = \sum_{i,j=1}^{1701} \nabla\mu_i \left(C^{-1}_{SN}\right)_{ij} \nabla \mu_j, \tag{28}
$$

Here, $p_j$ represents the free model parameters and $C_{SN}$ is the covariance matrix [69], and

$$
\nabla\mu_i = \mu^{th}(z_i, p_1, \ldots) - \mu_i^{obs}.
$$

where $\mu_{th}$ represents the value of the distance modulus predicted by our model, while $\mu_{obs}$ is its observed value.

Now, the $\chi^2$ function for the $H(z)$ + Pantheon + Analysis datasets is taken to be

$$
\chi^2_{total} = \chi^2_H + \chi^2_{SN} \tag{29}
$$

We present the $1\sigma$ and $2\sigma$ likelihood contours for the model parameters $\alpha$, $\gamma$, $\zeta$, and $H_0$, using the combined $H(z)$ + Pantheon + Analysis datasets, below.

The obtained best fit values from $1\sigma$ and $2\sigma$ contours presented in Figure 1, are $\alpha = 1.310^{+0.037}_{-0.032}$, $\gamma = 1.29 \pm 0.20$, $\zeta = 5.02 \pm 0.26$, and $H_0 = 72.09 \pm 0.19$.

Now we are going to present the cosmological implications of obtained observational constraints. We analyze the behavior of energy density, the pressure component incorporating viscosity, and the effective EoS parameter, for the obtained mean values of $H_0$ and model parameters $\alpha$, $\gamma$, and $\zeta$, constrained by the $H(z)$ + Pantheon + Analysis datasets.

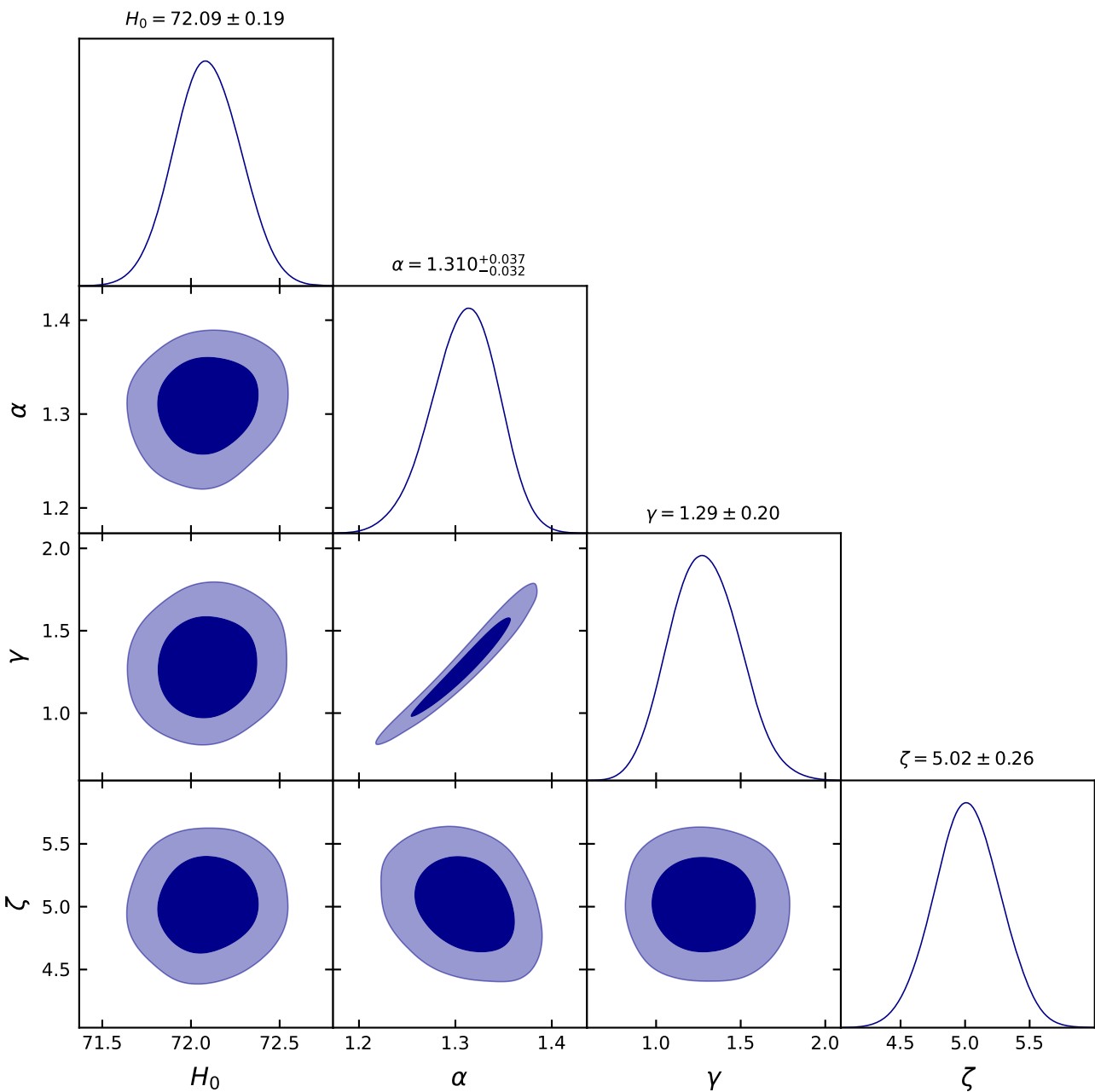

**Figure 1.** The $1\sigma$ and $2\sigma$ contours for the model parameters $\alpha$, $\gamma$, $\zeta$, and $H_0$, using the combined $H(z)$ + Pantheon + Analysis datasets.

We reconstructed the matter–energy density, the effective pressure, and the EoS parameter, as a function of the redshift, presented in Figures 2–4, for 7500 samples, that are reproduced by re-sampling the chains through *emcee*. From Figure 2, it is evident that the cosmic matter–energy density shows the expected positive behavior, and it vanishes with the expansion of the universe in the far future. The effective pressure component presented in Figure 3, exhibits negative behavior, that can lead to the accelerating expansion of the universe. Further, the present value of the effective EoS parameter is determined to be $\omega_0 \approx -0.71$. Thus, the behavior of the effective EoS parameter in Figure 4, confirms the accelerating nature of the expansion phase of the universe.

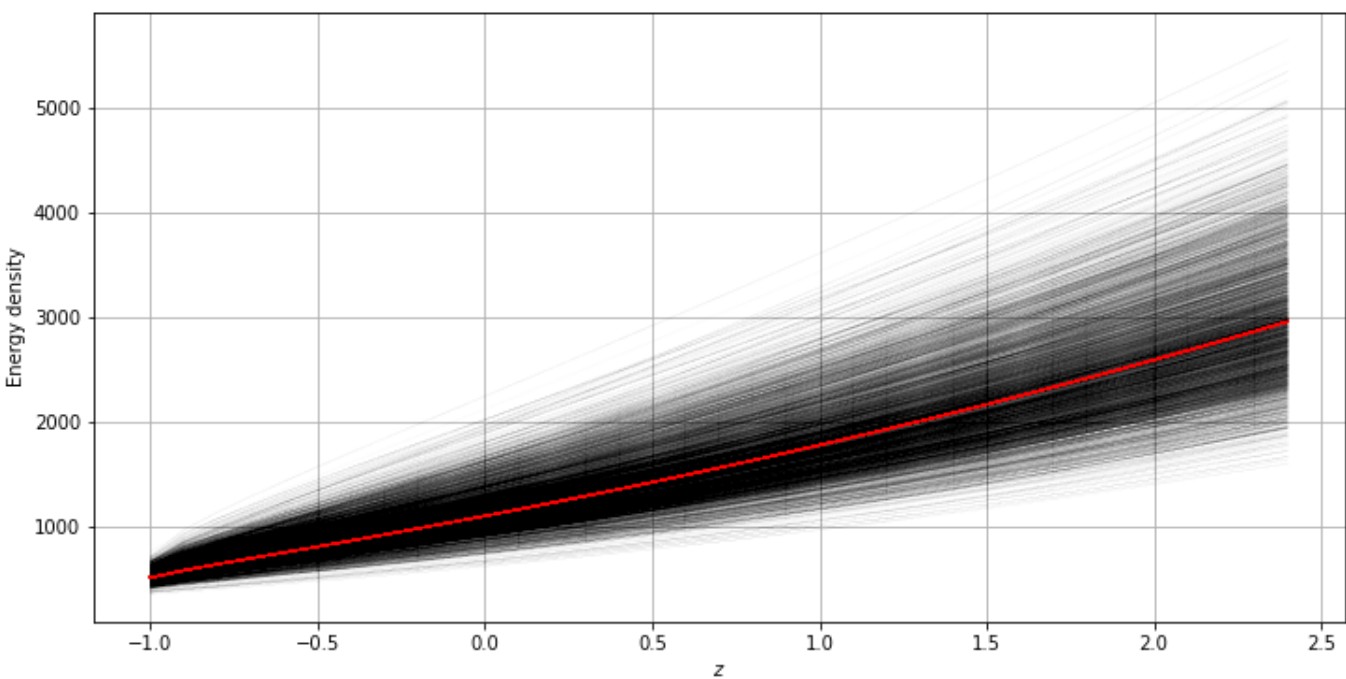

**Figure 2.** The reconstruction of the energy density as a function of the redshift, for our model, is presented for 7500 samples, which are reproduced by re-sampling the chains through *emcee*. We plot all the obtained curves, alongside the curve corresponding to the best fit of the parameters (red curve).

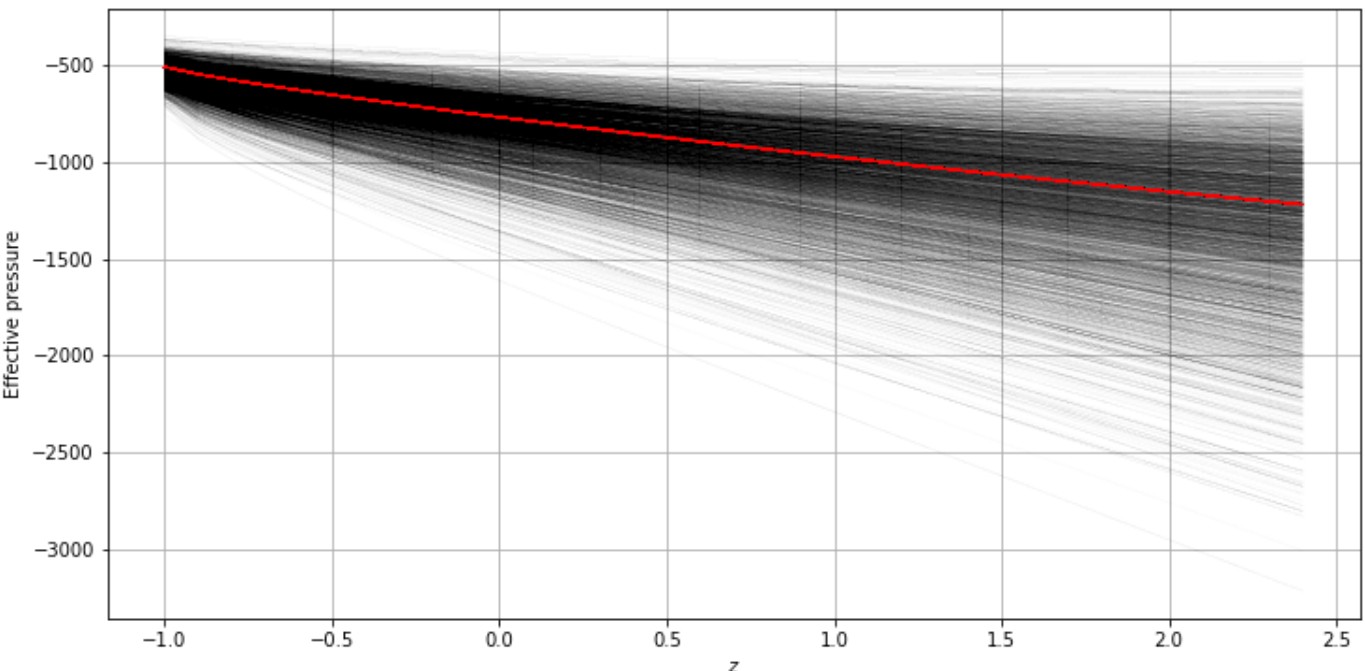

**Figure 3.** The reconstruction of the effective pressure as a function of the redshift, for our model, is presented for 7500 samples, which are reproduced by re-sampling the chains through *emcee*. We plot all the obtained curves, alongside the curve corresponding to the best fit of the parameters (red curve).

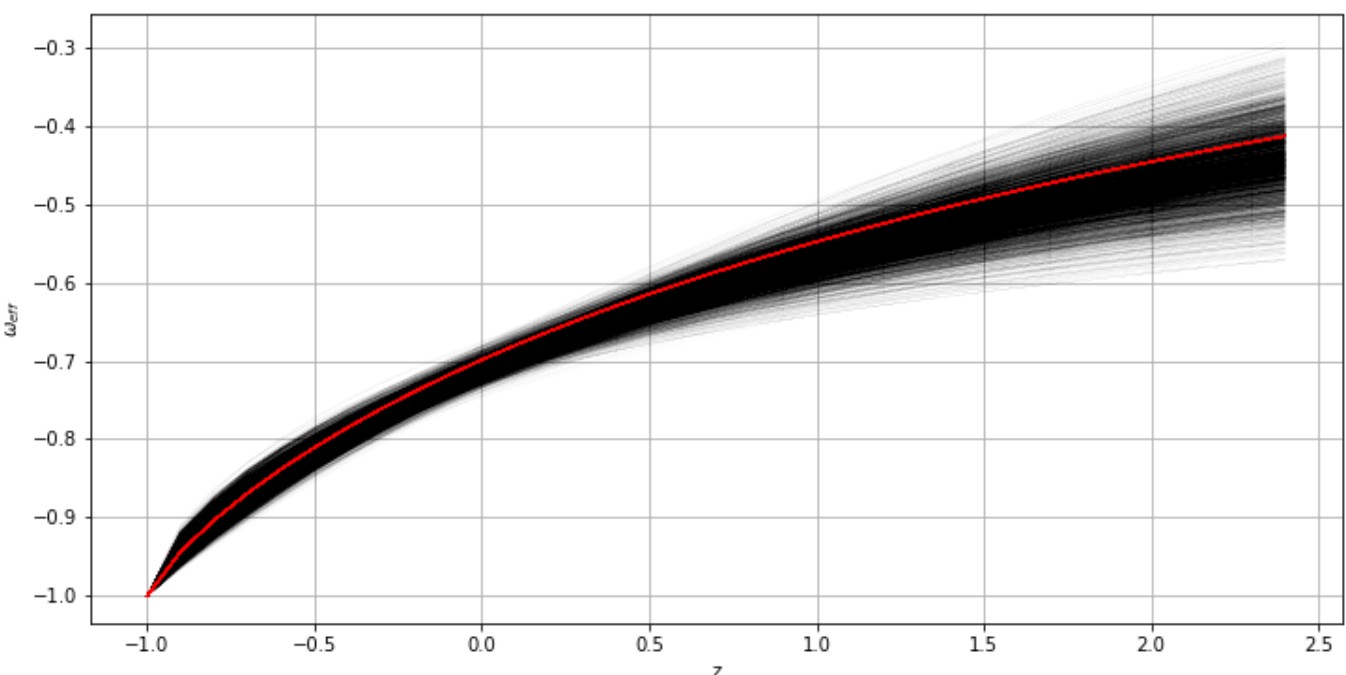

**Figure 4.** The reconstruction of the effective EoS parameter as a function of the redshift, for our model, is presented for 7500 samples, which are reproduced by re-sampling the chains through *emcee*. We plot all the obtained curves, alongside the curve corresponding to the best fit of the parameters (red curve).

## 6. Statefinder Diagnostic

It is well accepted that dark energy is responsible for cosmic expansion. In the last few decades, investigations into the origin and fundamental behavior of dark energy have increased. Consequently, plenty of dark energy models have started appearing, and therefore the distinction between these models of dark energy, either quantitative or qualitative, becomes necessary. In this direction, Sahni et al. [70] proposed a statefinder diagnostic method, that can discriminate between various dark energy models, with the help of a pair of geometrical parameters called statefinder parameters $(r, s)$. These are defined as

$$r = \frac{\dddot{a}}{aH^3} \tag{30}$$

and

$$s = \frac{(r-1)}{3(q - \frac{1}{2})} \tag{31}$$

We evaluate the statefinder parameters $(r, s)$ for our cosmological $f(R, L_m)$ model. The evolutionary trajectory of the assumed model, with the agreement of obtained observational constraints, is presented in Figure 5. The deviation of the evolutionary trajectory of the given model from the $\Lambda$CDM one, gives the required discrimination. The values $r = 1, s = 0$ represents the $\Lambda$CDM model, $r > 1, s < 0$ represents the Chaplygin gas model, and $r < 1, s > 0$ represents the quintessence model. The present value of statefinder parameters for our model is nearly $(r, s) = (0.43, 0.33)$. From Figure 5, it is evident that the dark component, due to modified geometry with the effect of bulk viscosity, has quintessence type behavior.

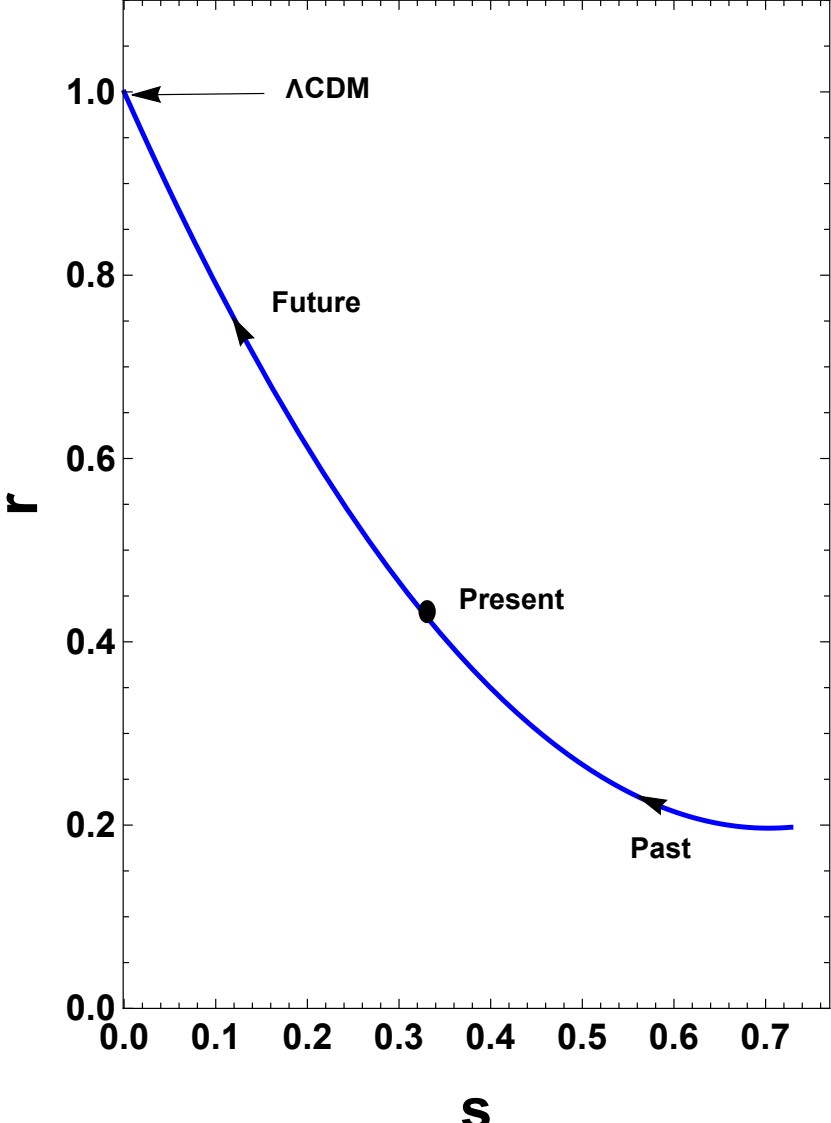

**Figure 5.** Profile of the evolution trajectory of the given model in the *r*–*s* plane, with the agreement of obtained observational constraints.

### 7. Om Diagnostics

The Om diagnostic is another recently proposed method that can effectively distinguish different dark energy models [71]. It is much simpler than the statefinder analysis, since it offers a formula incorporating only the Hubble parameter. For the spatially flat constraint, it is given by,

$$Om(z) = \frac{\left(\frac{H(z)}{H_0}\right)^2 - 1}{(1+z)^3 - 1} \qquad (32)$$

A negative slope of $Om(z)$, represents quintessence behavior, whereas a positive slope represents phantom behavior. The constant nature of $Om(z)$, corresponds to the $\Lambda$CDM type behavior of the given model.

Figure 6 indicates that the Om diagnostic parameter shows a negative slope in the entire domain. Thus, the Om diagnostic test indicates that our bulk viscous matter dominated $f(R, L_m)$ model follows the quintessence scenario.

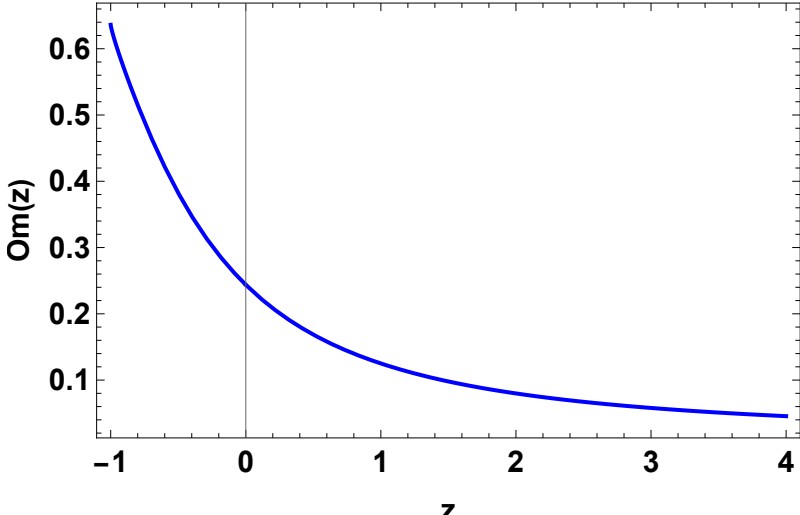

**Figure 6.** Profile of Om diagnostic parameter, with the agreement of obtained observational constraints.

## 8. Energy Conditions

Now, we are going to test the viability of the acquired solution corresponding to the assumed $f(R, L_m)$ model, by invoking the energy conditions criterion. The energy conditions, are criteria imposed to the energy–momentum tensor, in order to fulfill the positivity condition of energy. These criteria are offered from the excellent work of Raychaudhuri, that is known as Raychaudhuri's equation, and are written as [72]

- **Null energy condition (NEC):** $\rho_{eff} + p_{eff} \geq 0$;
- **Weak energy condition (WEC):** $\rho_{eff} \geq 0$ and $\rho_{eff} + p_{eff} \geq 0$;
- **Dominant energy condition (DEC):** $\rho_{eff} \pm p_{eff} \geq 0$;
- **Strong energy condition (SEC):** $\rho_{eff} + 3p_{eff} \geq 0$,

where $\rho_{eff}$ is the effective energy density.

From Figures 7 and 8, we observed that the NEC and DEC satisfy the positivity criteria in the entire domain of the redshift range, corresponding to the estimated values of parameters from the observational datasets. As WEC comprises energy density and NEC, it is also satisfied. Finally, Figure 9 shows that the violation of SEC occurs in the recent past, and hence this violation favors cosmic acceleration.

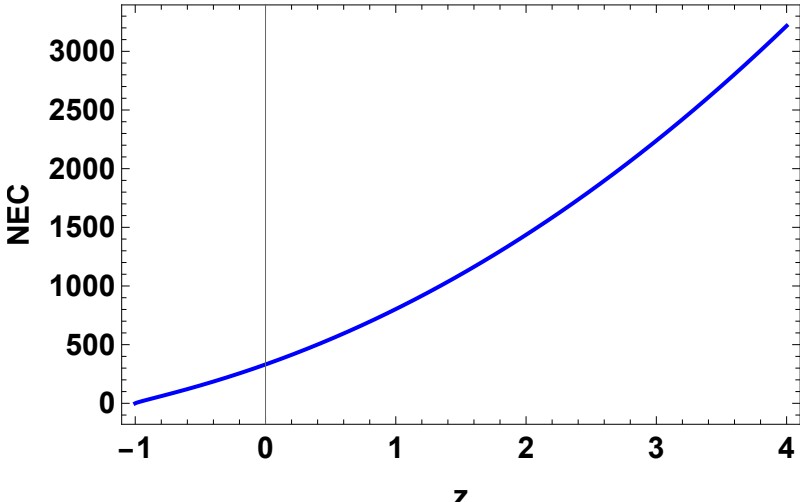

**Figure 7.** Behavior of the NEC vs. redshift.

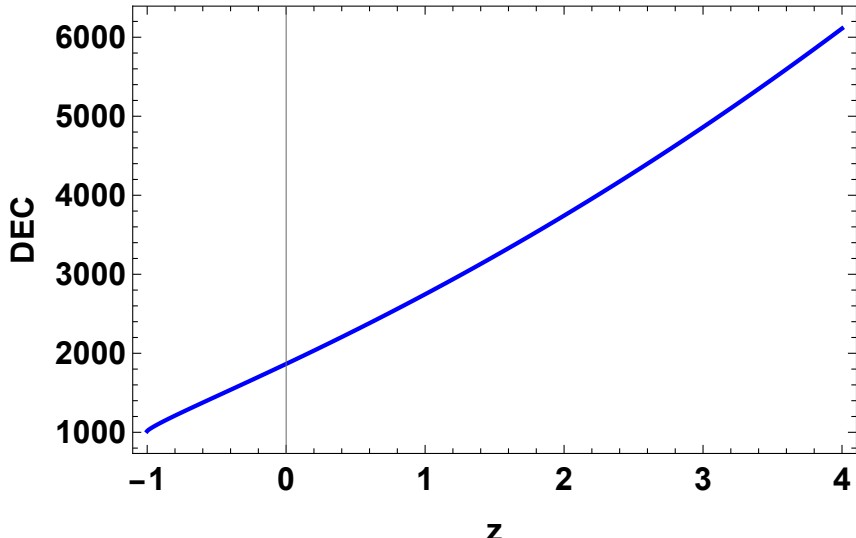

**Figure 8.** Behavior of the DEC vs. redshift.

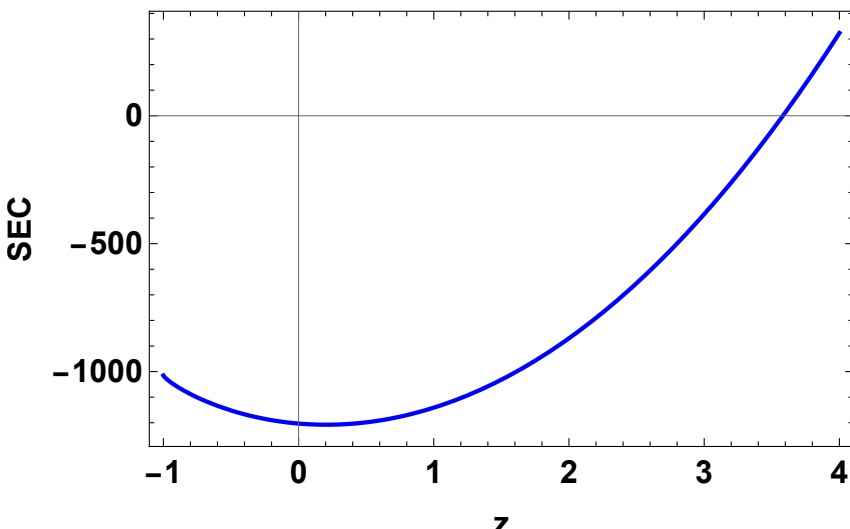

**Figure 9.** Behavior of the SEC vs. redshift.

## 9. Conclusions

Hydrodynamically, the inclusion of the coefficient of viscosity in the cosmic matter content is quite natural, as an ideal characteristic of a fluid is, after all, an abstraction. In the presented article, we have analyzed the significance of bulk viscosity in driving the cosmic late-time acceleration under the $f(R, L_m)$ background. $f(R, L_m)$ gravity theory is a generalization of matter–curvature coupling theories [39]. Harko and Lobo investigated curvature–matter couplings in modified gravity, from linear aspects to conformally invariant theories [32]. Models with non-minimal matter geometry couplings have great astrophysical and cosmological implications [35–38]. For our analysis, we considered an $f(R, L_m)$ function, particularly, $f(R, L_m) = \frac{R}{2} + L_m^\alpha$, where $\alpha$ is a free model parameter. Then, we assumed the effective equation of state in Equation (12), which is the Einstein case value, with proportionality constant $\zeta$, used in Einstein's theory [46], frequently used in the literature. We found the exact solution of our bulk viscous matter dominated $f(R, L_m)$ model, and then we used the combined $H(z)$ + Pantheon + Analysis observational datasets to constrain the present value of the Hubble parameter $H_0$ and the model parameters. The obtained best fit values are $\alpha = 1.310^{+0.037}_{-0.032}$, $\gamma = 1.29 \pm 0.20$, $\zeta = 5.02 \pm 0.26$, and $H_0 = 72.09 \pm 0.19$. In addition, we characterized the behavior of matter–energy den-

sity, a pressure component incorporating viscosity, and the effective EoS parameter, as a function of the redshift, presented in Figures 2–4, for 7500 samples, that are reproduced by re-sampling the chains through *emcee*. From Figure 2, it is evident that the cosmic matter–energy density shows the expected positive behavior, and the effective pressure component presented in Figure 3, exhibits negative behavior, that can lead to the accelerating expansion of the universe. Moreover, the present value of the effective EoS parameter is obtained to be $\omega_0 \approx -0.71$. Thus, the trajectory of the EoS parameter in Figure 4 confirms the accelerating nature of the expansion phase of the universe. Then, we evaluated the $(r, s)$ parameters for our assumed $f(R, L_m)$ model. The present values of the statefinder parameters for our model are nearly $(r, s) = (0.43, 0.33)$. In Figure 5, we observed that the evolutionary trajectory of our $f(R, L_m)$ model lies in the quintessence region. Further, the Om diagnostic, presented in Figure 6, indicates that our assumed $f(R, L_m)$ model favors the quintessence type dark energy. Finally, the energy conditions presented in Figures 7–9, exhibit positivity criteria in the entire domain of the redshift range corresponding to the case of NEC and DEC, whereas it shows the violation in the case of SEC. This violation of SEC, occuring in the recent past, favors the observed acceleration. We conclude that our cosmological $f(R, L_m)$ model, with the fluid incorporating the bulk viscosity effects, can efficiently interpret the late-time cosmic phenomenon of the universe with observational compatibility.

**Author Contributions:** L.V.J.: Writing—Original draft preparation, Investigation, Graph plotting, Methodology, Formal analysis; R.S.: Writing—Original draft preparation, Investigation, Graph plotting, Data curation, Methodology, Formal analysis; S.M.: Writing—Original draft preparation, Investigation, Graph plotting, Data curation; P.K.S.: Writing—Reviewing and Editing, Validation, Project administration, Supervision, funding. All authors have read and agreed to the published version of the manuscript.

**Funding:** L.V.J. acknowledges the University Grant Commission (UGC), Govt. of India, New Delhi, for awarding JRF (NTA ref. no.: 191620024300). R.S. acknowledges UGC, New Delhi, India for providing the Junior Research Fellowship (UGC-ref. no.: 191620096030). S.M. acknowledges the Department of Science and Technology (DST), Govt. of India, New Delhi, for awarding the Senior Research Fellowship (file no. DST/INSPIRE Fellowship/2018/IF18D676).

**Institutional Review Board Statement:** Not applicable.

**Informed Consent Statement:** Not applicable.

**Data Availability Statement:** There are no new data associated with this article.

**Acknowledgments:** We are very grateful to the honorable referees and to the editor, for the illuminating suggestions that have significantly improved our work in terms of research quality and presentation.

**Conflicts of Interest:** The authors declare no conflict of interest.

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
