# Peer review of "Constraining the Viscous Dark Energy Equation of State in f (R, Lm) Gravity"

_universe, doi:10.3390/universe9040163_

Round 1

Reviewer 1 Report

Everything is alright with the theoretical part, just need to provide details of \chi square analyses. How many observational parameters do they have? etc.

Author Response

File Attached

Reviewer 2 Report

Dear Editor and Authors,   In this manuscript, the $f(R,L_m)$ gravity is considered.
In this context, the importance of bulk viscosity to drive the cosmic l
ate time acceleration is analyzed. Using the combined
H(z) + Pantheon + BAO observational data, some parameters are found.
Constraints for the present value of the Hubble parameter are
discussed. The physical behavior of energy density, effective
pressure, and the effective EoS parameter are analyzed. The results
obtained with the theoretical model show compatibility with
observational data. I think the analysis and results presented
in the manuscript are correct and interesting. Furthermore, the
manuscript is well-written. Therefore, I think the paper may be
accepted, in its current form, for publication in Universe.

Author Response

Thanks for accepting our article.

Reviewer 3 Report

The authors introduce a new cosmological model based on f(R) modified gravity, where it also contains a power-law function of Lm. Moreover, they include a polytorpic fluid without explicitly adding baryonic and Dark Matter. Subsequently, they fit their model to the observational data to estimate the values of the free parameters for their model. 

Before suggesting the publication of their work, there are many questions to be answered.

Firstly, I would like to ask why the authors do not include standard matter and what is their motivation for including power-law Lm and in the same time an extra, polytropic fluid. How these assumptions are related to open question in the field, i.e. Hubble constant problem, \sigma 8 tension etc.?

Also, they cite arXiv:1407.2013 (Ref. 63), where a non minimal coupling is employed. However in the model cosider here, there is minimal coupling, as no coupling of R with L_m is present, i.e. eq. (15).

I am not against of the particular form of f(R,L_m) function used. However, more discussion could be helpful. The relevant discussion could be enriched with the aid of seminal works in the topic, i.e. arxiv: 0811.2876  and/or more recent ones, i.e. arxiv:2203.03295.

Secondly, they claim right after eq. (15) that their model goes to GR limit for a = 1.

This is wrong, and the fact that a = 0 do not provide GR limit can be easily deduced by setting a = 1 at eq. (21). By doing this, we do not end up with H \sim H_0 (1+z)^(3/2) as expected for the case of GR. Instead, we need also \zeta = 0 and \gamma = 1. 

Third, the four free parameters of their model are not independent, as theey are subject to a normalization condition (definition of H_0) must be satisfied at any time (or z), i.e H(z=0) = H_0 which corresponds to g(\zeta,gamma,\alpha) = 0 and via this expression the free parameters of the model reduce to 3. Fitting all 4 free parameters, namely \alpha, \gamma, \zeta and H_0 is wrong, as the normalization relation is not necessarily satisfied.

There also a number of important technical points, i.e. about half of the 57 H(z) data considered are not cosmological model - independent (i.e. p. 8 of arxiv:1507.08279) and also the BAOs fitting includes some cosmological model dependent assumptions, i.e. the standard ruler, r_z, (eq (29) is only valid for LCDM, check etc. the Appendix of arxiv:1806.10580 and corresponding references). Moreover, as a suggestion, it would be better to use the latest SN Ia data, i.e. Pantheon+ data arxiv:2112.03863.

Finally, (after the above) there are some other improvements, i.e. provide the reconstructed w(z) of the model (by resampling the mcmc chains).

There exist a number of minor points, while part of them are the following. The cited refs [3],[4] right after "... plethora of observational results such as Large Scale Structure ..." in the Introduction describe ways to check modified cosmologies 

via LSS. A review paper on LSS surveys could be more apropriate in this place. The analogous holds for Refs [6,7] and Ref. 10, i.e. consider using the PLANCK paper arxiv:1807.06209 (also for the provided equation of state parameter value). All the aforementioned Refs that contain work on f(R,L_m) models should be included in the discussion about these models. I would also kindly ask for providing directly the links (or arxiv numbers) for any referred papers in your reply (if any).

Author Response

File attached

Round 2

Reviewer 3 Report

The authors incorporated some of the modifications requested in the updated version of the article.
However, they do not added the constrain of the normalization of the Hubble equation, i.e.
i.e H(z=0) = H_0 which corresponds to g(\zeta,gamma,\alpha) = 0. The importance of this is enhanced by the fact that
they found an increased H_0 value, closer to the Riess et al result than PLANCK's. I am very curious of the relevant result, as it could have implications regarding the H_0 tension, so i would like to see the aforementioned modification.

Moreover, i would like to ask why they do not use BAOs data.
Another suggestion is to use the extracted values for the free parameters in order to add errors at Fig. 2,3,4, in the same manner with i.e. Fig. 5 of arxiv: 2012.06524.
